# Outbreaks of COVID-19 in Nursing Homes: A Cross-Sectional Survey of 74 Nursing Homes in a French Area

**DOI:** 10.3390/jcm10184280

**Published:** 2021-09-21

**Authors:** Emilie Piet, Alexis Maillard, Franck Olivier Mallaval, Jean Yves Dusseau, Murielle Galas-Haddad, Sébastien Ducki, Hélène Creton, Marc Lallemant, Emmanuel Forestier, Gaëtan Gavazzi, Tristan Delory

**Affiliations:** 1Centre Hospitalier Annecy Genevois, 74330 Epagny Metz-Tessy, France; mgalashaddad@ch-annecygenevois.fr (M.G.-H.); hcreton@ch-annecygenevois.fr (H.C.); tdelory@ch-annecygenevois.fr (T.D.); 2The PROCOPAD Study Group, 74330 Epagny Metz-Tessy, France; maillard.alexis@laposte.net (A.M.); franckolivier.Mallaval@ch-metropole-savoie.fr (F.O.M.); JYDusseau@ch-alpes-leman.fr (J.Y.D.); SDucki@chu-grenoble.fr (S.D.); marclallemant@gmail.com (M.L.); Emmanuel.Forestier@ch-metropole-savoie.fr (E.F.); GGavazzi@chu-grenoble.fr (G.G.); 3Centre Hospitalier Métropole Savoie, 73000 Chambéry, France; 4Centre Hospitalier Alpes Léman, 74130 Contamine sur Arve, France; 5Centre Hospitalier Universitaire Grenoble Alpes, 38700 La Tronche, France; 6Faculty of Associated Medical Sciences, Chiang Mai University, Chiang Mai 50000, Thailand

**Keywords:** nursing homes, outbreak, COVID 19, staff

## Abstract

In this multi-centric cross-sectional survey conducted in nursing homes of the French Alps, from 1 March to 31 May 2020, we analyze the relationship between the occurrence of an outbreak of COVID 19 among residents and staff members. Out of 225 eligible nursing homes, 74 (32.8%) completed the survey. Among 5891 residents, the incidence of confirmed or probable COVID-19 was 8.2% (95CI, 7.5% to 8.9%), and 22 (29.7%) facilities had an outbreak with at least 3 cases. Among the 4652 staff members, the incidence of confirmed or probable COVID-19 was 6.3% (95CI, 5.6% to 7.1%). A strong positive correlation existed between residents and staff members for both numbers of cases (*r*^2^ = 0.77, *p* < 0.001) and the incidence (*r*^2^ = 0.76, *p* < 0.001). In univariate analyses, cases among the staff were the only factor associated with the occurrence of an outbreak among residents (OR = 11.2 (95CI, 2.25 to 53.6)). In bivariate analysis, this relationship was not influenced by any nursing home characteristics, nor the action they implemented to mitigate the COVID-19 crisis. Staff members were, therefore, likely to be a source of contamination and spread of COVID-19 among nursing home residents during the first wave of the pandemic.

## 1. Introduction

The COVID-19 pandemic state was declared by the World Health Organization (WHO) on 11 March 2020. In Europe, the first wave of the pandemic was associated with an excess mortality of nearly 200,000 deaths over 8 weeks, of which 91% occurred among people ≥ 65 years of age [1]. In France, the infection fatality ratio was 0.5%, ranging from 0.001% under 20 years of age to 8.3% over 80 years of age [2].

In nursing homes, the morbidity associated with COVID-19 was concentrated in those with documented outbreaks [3]. Mortality was higher among residents of nursing homes than among elderly living in independent housing [4]. Aside the poor underlying condition of nursing home residents, it was suspected that the higher mortality observed was related to outbreaks initiated by care workers and visitors [4]. Shah and colleagues showed that care workers, along with their households, contributed to a sixth of hospital admissions, and the likelihood of hospitalization was increased in care workers facing patients [5]. However, to date, the existence of a reversed relationship for residents facing care givers has not been documented in nursing homes. Importantly, little is known about the ways the virus enters and spreads within this complex environment, a space where people live and where they receive care. Strict observance of hygienic principles have been re-emphasized in the last guidelines of the European Geriatric Medicine Society and by the US Center for Disease Control, resulting in specific recommendations for nursing home residents, staff and visitors to prevent COVID-19 outbreaks [6,7].

In France, during the first wave, a total of 5203 outbreaks (≥1 case) was reported in nursing homes. In the Auvergne-Rhône-Alpes region, where 651 occurred, 3885 residents had confirmed COVID-19 infection, and 1772 (46%) subsequently died [8]. To mitigate the health crisis, the French government promulgated a ban on visits to residents and the reinforcement of hygiene measures targeting care workers. This decision was followed by a national lockdown.

We aim to report the correlation between the occurrence of an outbreak among nursing home residents and staff members, and explore the factors influencing this relationship, among nursing homes of the French Alps, a part of the Auvergne-Rhône-Alpes region.

## 2. Materials and Methods

### 2.1. Study Context and Design

The French Alps, a part of the Auvergne-Rhône-Alpes region, group 2,497,256 inhabitants over three departments of France: Isère, Savoie and Haute Savoie. It is a semi-rural area where the first French outbreaks of COVID-19 cases were reported [9].

In France, private and public nursing homes are equally supported by the national health insurance system for care. Private facilities charge residents a larger share of the cost of housing than public facilities. Public nursing homes can be part of a public hospital network, while private ones are always independent of hospitals. In the Auvergne-Rhône-Alpes area, a nursing home can contract an agreement with infection control teams from public hospitals through a partnership with the regional health authorities. These teams are independent of nursing home direction. They include an infection control practitioner and infection control nurses. One team oversees 13 to 45 nursing homes. They support the facilities to formalize their infectious risk prevention policy, train the staff members, and intervene on-site in case of an outbreak.

Between 15 July and 15 November 2020, we conducted an online cross-sectional survey within the 225 nursing homes of the French Alps to investigate retrospectively the first wave of the COVID-19 pandemic, from 1 March to 31 May 2020.

The ban on visits to residents was issued on 11 March, followed by the national lockdown from 17 March to 11 May. Since the first wave, directors have to report to regional health authorities the daily number of cases among the residents and the personnel. It was estimated that after the first wave, the prevalence of COVID-19 was of 4.4% (95CI, 2.8–6.5%), ranging between 1.7% in those ≥70 years of age, and 6.3% in those <40 years of age [10]. In a recent survey that we conducted among 3454 healthcare workers from four public hospitals of the French Alps, the overall prevalence of SARS-CoV-2 infection was 5.0% (95CI, 4.3% to 5.8%) [11]. No data about local incidence during the first wave are available.

The study questionnaire was developed by the ProCoPAD Study group, including infectious diseases, hygiene and geriatric specialists. It was tested and validated by directors of nursing home in a pilot phase. It was made available on the internet 24 h/day, 7 days/week using the dedicated link sent to the directors of each institution through email. Monthly reminders (email and phone) were sent out until 7 November 2020.

### 2.2. Content of the Survey

The survey documented in an aggregated fashion the nursing home characteristics: type (public or private institution), number of residents, number of dementia beds, the average scores for dependency and morbidity of the residents (detailed in Appendix A) [12,13,14]. Institutions were asked to report on the human resources used during the epidemic, as well as their operational management of the crisis: number of personnel, use of on-demand manpower, date of effective visit ban, face mask wearing conditions for personnel, and residents’ confinement in their rooms. The agreement with an infection control team was sought. Finally, epidemiological data extracted from the official registry of the nursing homes was collected: number of confirmed, probable or possible cases of COVID-19, as well as hospitalizations and deaths among residents, turnover of residents, and number of probable or confirmed cases among the personnel. Questionnaire details are provided in Appendix A.

#### 2.2.1. COVID-19 Cases and Populations of Analysis

French Health Authorities define COVID-19 cases using a classification system similar to that of WHO [15]. A [confirmed case] was a person (resident or staff) with a positive biological test (RT-PCR or serology) confirming the SARS-CoV-2 infection. A (probable case) was a person (resident or staff) with clinical signs and thoracic CT-scanner compatible with COVID-19. A (possible case) was a resident with physical signs compatible with influenza-like illness or COVID-19, without radiological or biological confirmation [16,17].

For our main analysis, we used subjects with (confirmed or probable) COVID-19 as cases among the nursing home residents and staff members. In sensitivity analyses we used confirmed infections for residents and staff members only, or the three categories lumped together for the residents.

#### 2.2.2. Outbreaks

An outbreak was said to have occurred if at least three (confirmed or probable cases) had been identified among residents over the 8 weeks study period. In sensitivity analyses we assumed that an outbreak had occurred when at least one confirmed or probable case had been identified in a resident. This was justified by the up to 50% proportion of asymptomatic cases reported among elderly in care institutions, and because access to RT-PCR, serology, and CT scan was poor during the first wave of the pandemic, resulting in an underestimation in confirmed and probable cases [2,18].

### 2.3. Statistical Analyses

We computed frequencies and percentages for discrete variables, and the median and interquartile range (IQR) for continuous variables. We used the chi-squared test to compare percentages. The incidence of COVID-19 and its 95% confidence interval (95CI) was measured as the number of cases over the number of exposed (residents or staff members) during the 8 weeks duration of the study. To investigate the relationship between residents and staff members for the occurrence of outbreaks and their spread, we computed the linear coefficient of determination (r-squared, *r*^2^) between the number of cases among residents and the number of cases among staff members, and between the incidence of COVID-19 among residents and that observed among staff members. In sensitivity analyses, we investigated the persistence of these correlations using different case definitions.

We then used logistic regressions to investigate the strength of associations between the occurrences of outbreaks in nursing home (outcome) and explanatory variables including nursing home characteristics or action implemented to curb the epidemic, adjusted on the occurrence of cases among staff members (Model A). Associations are reported as odds ratio (OR) and their 95%CI. All tests were two-tailed, and the level of significance was set at 5% bilaterally. Finally, we conducted a set of sensitivity analysis: imputation of missing data by multiple chained equations (Model B), outbreak ≥ 1 cases among residents without and with imputation of missing data (Model C and Model D). Analyses were performed on R, version 4.0.1 (R Foundation for Statistical Computing, Vienna, Austria).

### 2.4. Ethics, Patient and Public Involvement

No patients were involved in setting the research question or the outcome measures, nor were they involved in developing plans for design of the study. No patients were asked to advise on interpretation or writing up of results.

In accordance with French regulation, this study was not approved by an institutional review board as the data collected were aggregated (Article R1121-1 of the French Public Health Code). None of the collected data are shared with private companies. Confidentiality policy fulfils the European General Data Protection Regulation.

## 3. Results

### 3.1. Nursing Homes Characteristics and Measures to Control the Epidemic

Of the 225 nursing homes invited to participate, 74 (32.8%) completed the online questionnaire, accounting for 5891 residents and 4652 staff members (Appendix A). The median number of residents in the respondent nursing homes was 74 [60 to 82], with a median dependency score, and median morbidity score comparable to that reported in the survey area (see Appendix A). A third (35.1%) were private care facilities, with a median number of caregiver per 100-residents of 43.5 [39.0 to 51.7]. Most (86.4%) of the nursing homes recruited additional manpower (interim, assistance from other institutions, and volunteers) during the health crisis. Visits to residents were effectively banned in all facilities after a median of 0 days [−2.5 to 1.5] from the national state decision. In all institutions, caregivers had the obligation to wear a face mask, a decision implemented a median of 6 days before [−15.2 to 0.2] national lockdown. Masks were to be replaced daily (40.5% of institutions) or every 4 h (59.5%). Only one nursing home ran out of hydro-alcoholic solution.

Residents were confined to their rooms in 86.1% of the nursing homes. Almost all facilities (95.9%) were associated with an infection control team, which intervened in 91.7% of the cases. A screening of all staff members for COVID-19 by nasal RT-PCR was organized in 63.5% of nursing homes. The nursing home characteristics and measures implemented to prevent and control the epidemic are detailed in Table 1.

### 3.2. COVID-19 among Residents and Staff Members

Table 2 describes cases among residents and staff members by the occurrence of outbreaks among residents.

Among 5891 residents, the incidence of confirmed or probable COVID-19 was 8.2% (95CI, 7.5% to 8.9%). As a result, 22 (29.7%) nursing homes had an outbreak, and 29 (39.2%) had at least one confirmed or probable case of COVID-19. Overall, 586 residents (10.0% (95CI, 9.2 to 10.8)) were hospitalized, and 503 (8.5% (95CI, 7.8 to 9.2)) died from any cause during the first wave. The cases of COVID-19, as well as hospitalization and death of all-cause were concentrated in nursing home with outbreaks of COVID-19 (*p* < 0.001). Among the 4652 staff members, the incidence of confirmed or probable COVID-19 was 6.3% (95CI, 5.6% to 7.1%), and 58.6% of nursing homes reported at least one COVID-19 case among staff. Cases among the staff members were also concentrated in institutions with outbreaks (18.5% versus 1.4%, *p* < 0.001).

### 3.3. Relationship between Infections among Staff Members and Residents

A strong positive correlation existed between residents and staff members for both the number of COVID-19 cases (*r*^2^ = 0.77, *p* < 0.001) and their incidences (*r*^2^ = 0.78, *p* < 0.001) (Figure 1). Results were similar in sensitivity analysis (Appendix A).

All but four nursing homes with COVID-19 cases among their residents reported cases among their staff members (85.1% (95CI, 66.2% to 95.8%)), while half (43.1% (95CI, 28.3% to 59.0%)) of nursing homes without cases among residents reported cases among staff members (Figure 2A and Appendix A).

Half (54.7% (95CI, 38.7% to 70.1%)) of nursing homes with infected staff members also reported cases among residents, but almost none of those COVID-19 free among their staff reported infection among their residents (13.8% (95CI, 3.9% to 31.7%)) (Figure 2B and Appendix A).

Furthermore, in facilities with outbreaks among residents when staff were infected, we observed a higher likelihood of COVID-19 infection (OR = 4.2 (95CI, 2.6 to 6.8)), hospitalization (OR = 2.8 (95CI, 1.7 to 4.7)) and death (OR = 2.2 (95CI, 1.3 to 3.7)) (Appendix A).

### 3.4. Nursing Homes Organization and Occurrence of Outbreaks among Residents

In univariate analyses, cases among the staff were the only factor associated with the occurrence of an outbreak among residents (OR = 11.2 (95CI, 2.3 to 53.6)) (Table 3). In bivariate analysis, this relationship was not influenced by any of the nursing home’s characteristics, nor the actions they implemented to mitigate the COVID-19 crisis (Table 3). Similar results were found in the sensitivity analysis using multiple imputation for missing data and case or outbreak definition (Appendix A).

## 4. Discussion

During the first wave of the COVID-19 pandemic, the incidence of infection was 8.2% (95CI, 7.5% to 8.9%) in residents of nursing homes of the French Alps, and 6.3% (95CI, 5.6% to 7.1%) in its staff. The occurrence of cases in residents and staff was concentrated in nursing homes with outbreaks. A strong positive correlation existed between residents and staff members for both the number of cases (*r*^2^ = 0.77, *p* < 0.001) and the incidences (*r*^2^ = 0.78, *p* < 0.001). The relationship was asymmetric, and cases within the staff members was associated to a five- to 11-fold increase in cases among the residents.

The characteristics of nursing homes in our sample, including size and residents’ average score of dependence and morbidity, was consistent with value reported for the surveyed area. The 8.2% incidence of COVID-19 in nursing homes was higher than in the general population of the surveyed area (4.4%) and in France (4.9%), even when stratifying on those ≥70 years of age (from 1.7% in in Auvergne-Rhône-Alpes, to 4.5% in France), but much lower than in United States nursing homes (28.4%) [10,19]. This higher incidence is likely to be driven by the frailty and underlying conditions of nursing home residents, and the close living environment which may favor viral spread [4,20]. We observed that, as in Italy, ~30% of nursing homes faced an outbreak [21]. The hospitalization rate among residents (10% overall and 19% in facilities with outbreak) was lower than among the elderly population living in household (up to 38%) but similar to Italian reports for nursing homes [2,21]. The contribution of geriatric and palliative care physicians could have avoided transfers to hospitals. The all-cause mortality among residents (8.5% overall and 14.1% in facilities with outbreak) was lower than in elderly admitted from the community to hospital for COVID-19 (20% to 38%), but similar to the mortality rate observed in France for nursing homes (9.5%) [22,23].

In conditions where visits were banned and most of the residents were confined in their rooms, the relationship between COVID-19 cases among staff members and residents was not symmetric. The staff members were likely to introduce (Figure 2A,B) and spread the virus. Indeed, if infection among staff members was associated with COVID-19 among residents in about half of the cases, only very few cases in residents occurred with no reported infection among staff members. This was highly suggestive that the virus is introduced in nursing homes by staff members. Once introduced, the virus spread widely, with incidences as high as 40% and associated to a substantial morbidity. As a comparison, in French nursing homes, outbreaks of influenza-like illness usually have an incidence of 25% to 30% but result in hospitalization in only 6% to 9% of the cases, and death in 2% to 3% [24]. Importantly, in the case of outbreak among residents, COVID-19 incidence was significantly higher if the outbreak involved staff members, which is strongly suggestive of a virus spreading by staff members (Appendix A). As a consequence, hospitalization rate and mortality were also significantly higher when outbreaks involved staff.

The introduction and the spread occurred despite the visit ban, national lockdown, an extensive access to infection control team, and regardless of nursing home characteristics. The actions implemented did not curb the epidemics but their scaling-up was quite uniform in our study. Discrepant reports were compiled in the United States and United Kingdom about the influence of facilities’ quality rating on infection rate [25,26,27]. Such quality scales are not used in French nursing homes, and we thereby cannot provide insight regarding this potential confounder. Other studies have shown that a high number of beds or lower staffing ratios were associated with the occurrence of outbreaks [3,21,28,29]. However, in these studies incident cases among staff members were not accounted for. This underlines the need to focus preventive action on staff members, and likely visitors: vaccination, reinforcement of hygiene measures, active screening of cases, and home confinement in case of positive testing of a caregiver [6,7,30]. In a recent study it was estimated that up to 22% of French healthcare workers were not willing at all to be vaccinated against COVID-19, and that only 30% would be likely to accept a mandatory immunization program [30]. Tools such as the Haddon matrix used in the injury prevention field and recently for COVID-19 in nursing homes, or the CDC’s Infection Control Assessment and Response (ICAR) tool are thereby of importance both at national and facility levels, in order to assess the feasibility and assist decision making in the prevention and containment of COVID-19 outbreaks in nursing homes [31,32].

Our study has limitations. First, it was a cross-sectional survey where only a third of eligible facilities responded. Nursing homes who responded might differ from those who did not. However, the nursing homes characteristics of the respondents were similar to those reported for the French Alps. Also, hospitalizations and mortality rates were comparable to other studies conducted in nursing homes and to national data [21,23]. Second, the ban of visits occurred 7 days before the studied period, and we cannot rule out the possibility that visitors introduced the virus prior to the ban. Incidences may have been underestimated as there was no systematic screening, and PCR testing was prompted by the occurrence of symptoms [2,18]. Thus, it was not possible to compare the timeline between detection of cases among residents and staff members. Comparison of incidence and mortality in nursing homes and in the elderly population receiving care at home is difficult as the two populations and their environments are not comparable. Also, it was difficult to compare our results with those from other reports as eligibility for care in nursing homes, their organization and level of medical oversight differ greatly within and across countries. Moreover, to estimate incidences we used a conservative approach based on the total number of residents present in nursing homes during the surveyed period. Other reports base their estimation on number of occupied bed at baseline, which leads to inflated estimates [33]. We were not able to explore the effect of the ethnicity of facilities personnel and their residents on incidence and mortality, while in the United States ethnicity was associated with the likelihood of COVID-19 cases and death [34]. Nursing homes had quasi-universal access to an infection control team, so we were not able to estimate the effect of preventive recommendations or measures on the occurrence of an outbreak nor on its intensity. As data were aggregated, we could not estimate the individual effect of collective measures. We were not able to evaluate the effect of personal protective equipment shortage as all had access to face masks and only one ran out of hydro-alcoholic solution. The efficacy of actions such as confining of residents in their rooms or systematic screening of staff members could not be assessed. Finally, it was not possible to investigate study transmission between visitors and residents. In a comprehensive approach to protect residents, similar actions to that implemented for staff members will thereby have to be conducted for visitors.

## 5. Conclusions

During first wave of national lockdown and visit ban in nursing homes, staff members were likely to be a source of viral contamination and spread among residents. None of the characteristics, organization or hygiene measures implemented by facilities seemed to reduce the occurrence of outbreaks. Measures to limit the incidence of COVID-19 in the general population and strong preventive actions, including vaccination, targeting staff members and visitors, are needed to limit the risk of further epidemics in nursing homes.

## Figures and Tables

**Figure 1 jcm-10-04280-f001:**
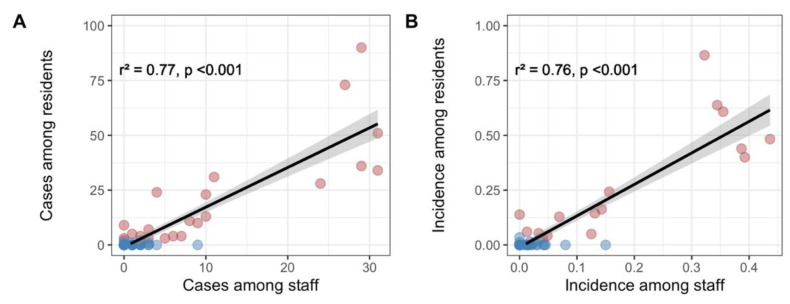
Linear correlations between number of cases and incidence of COVID-19 among the residents and staff members of nursing homes. The linear coefficient of determination (*r*^2^) between cases (**panel A**) and incidence (**panel B**) among residents and staff members are measured within the population of confirmed or probable cases. The red dots are representing nursing homes in which outbreaks (≥3 cases) among the residents occurred, while blue dots are representing nursing homes free of outbreak. The black line represents the linear regression line between cases in residents and in personnel, and the grey area its 95% confidence interval.

**Figure 2 jcm-10-04280-f002:**
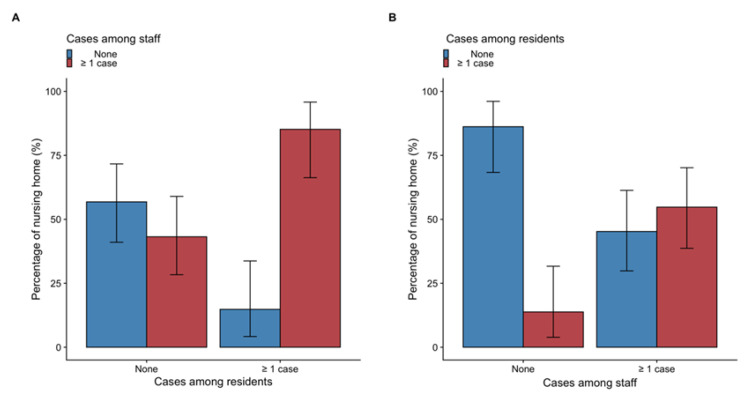
Relationship between COVID-19 cases either in nursing home residents or staff members and the occurrence of case among staff members or the residents. (**Panel A**) represents the frequency of nursing homes with ≥1 cases among its staff members according to the occurrence of ≥1 cases among its residents. (**Panel B**) represents the frequency of nursing homes with ≥1 cases among the residents according to the occurrence of ≥1 cases among its staff. Very few nursing homes had cases among their residents without having at least one case among staff members.

**Table 1 jcm-10-04280-t001:** Characteristics of nursing homes (NHs) and of the measures implemented to prevent and control the epidemic overall, in outbreak and in outbreak-free nursing homes.

NHs’ Characteristics, *n* (%) or Median [IQR]	All NH *n* = 74	Outbreak-Free NH *n* = 51	Outbreak NH * *n* = 22
Presence of coordinating physician	63 (85.1%)	43 (82.7%)	20 (90.9%)
Private nursing home	26 (35.1%)	14 (26.9%)	12 (54.5%)
NHs’ status			
Private associative NHs	18 (24.7%)	13 (25.0%)	5 (23.8%)
Private commercial NHs	5 (6.8%%)	1 (1.9%)	4 (19.0%)
Public hospital NHs	28 (38.4%)	22 (42.3%)	6 (28.6%)
Public territorial NHs	20 (27.4%)	16 (30.8%)	4 (19.0%)
Others	2 (2.7%)	0 (0.0%)	2 (9.5%)
Agreement with an infection control team	71 (95.9%)	50 (96.2%)	21 (95.5%)
Number of residents ^a^	74.5 [60.2, 83.5]	71.0 [54.8, 82.5]	77.5 [72.0, 83.5]
Number of dementia places	14.0 [0.0, 24.0]	12.5 [0.0, 22.0]	14.0 [2.0, 26.0]
Twin bedrooms ≥ 1	22 (30.1%)	15 (29.4%)	7 (31.8%)
Turnover rate ^b^	2.5 [0.0, 6.4]	2.5 [0.0, 7.2]	2.0 [0.0, 4.8]
Morbidity score, as per GMP ^c^	780.0 [746.8, 819.0]	783.0 [746.8, 818.8]	776.5 [751.0, 820.8]
Dependence score, as per PMP ^d^	227.0 [206.0, 244.0]	230.0 [206.2, 248.8]	219.0 [201.0, 239.5]
Number of staff ^e^	65.0 [48.0, 78.5]	64.0 [48.0, 76.5]	67.5 [54.5, 80.8]
Number of fulltime caregiver per 100 residents ^e^			
Nurse	4.4 [3.6, 6.0]	4.3 [3.2, 6.0]	4.9 [4.0, 6.0]
Assistant nurse	16.0 [11.5, 21.8]	16.0 [9.4, 22.0]	16.0 [12.0, 19.8]
Other caregivers	10.0 [7.0, 15.0]	10.5 [7.1, 16.0]	10.0 [7.0, 13.0]
Use of interim manpower	49 (68.1%)	36 (72.0%)	13 (59.1%)
Use of other sources of on-demand manpower	57 (78.1%)	38 (74.5%)	19 (86.4%)
Preventive measures implemented			
Days between national decision and visits’ ban	0 days [−2.5, 1.5]	0.5 days [−2.8, 2.0]	0 days [−2.0, 1.0]
Days between national lockdown and systematic face mask wear	−6.0 days [−15.2, 0.2]	−6.5 days [−17.5, 1.0]	−5.0 days [−13.2, 0.0]
Replacement of the facial mask every 4 h	44 (59.5%)	31 (59.6%)	13 (59.1%)
Individual lockdown of residents’ in their room	62 (86.1%)	42 (82.4%)	20 (95.2%)
Corrective measures implemented			
Intervention of the infection control team	65 (91.5%)	43 (87.8%)	22 (100.0%)
Systematic screening of all staff using RT-PCR for COVID-19	46 (63.0%)	28 (54.9%)	18 (81.8%)

* Outbreaks are defined as the occurrence of ≥3 cases among the residents of nursing home. ^a^ The number of residents was measured at time of study start: 1 March 2020. ^b^ The turnover rate was measured as the number of residents newly admitted over the number of beds available during the study period. ^c^ GMP: “Groupe iso-ressources Moyen Pondéré” is a score assessing the average level of dependency of nursing home residents. The GMP scale ranges from 70 points (very low level) to 1000 points (very high level). ^d^ PMP: “Pathos Moyen Pondéré”, is a score assessing the average level of morbidity of the nursing home residents. The scale ranges from 180 (very low level) to 260 (very high level). ^e^ Measured for the 8-week study period only.

**Table 2 jcm-10-04280-t002:** Number of cases and incidence of COVID-19 among residents and staff members, overall, in outbreak and in outbreak-free nursing homes.

	All NH *n* = 74	Outbreak-Free NH *n* = 51	Outbreak NH * *n* = 22	*p*-Value
**Residents**				
Number of residents	5891	4096	1795	
Incidence of COVID-19 among residents	482 (8.2%)	9 (0.2%)	473 (26.4%)	<0.001
Hospitalization rate	588 (10.0%)	247 (6.0%)	341 (19.0%)	<0.001
Mortality rate	503 (8.5%)	250 (6.1%)	253 (14.1%)	<0.001
**Staff**				
Number of staff	4652	3304	1348	
Incidence of COVID-19 among staff members	296 (6.4%)	46 (1.4%)	250 (18.5%)	<0.001
Facilities with ≥1cases among staff members	42 (56.8%)	23 (44.2%)	19 (86.4%)	<0.001

* Outbreaks are defined as the occurrence of ≥3 cases among the residents of nursing home.

**Table 3 jcm-10-04280-t003:** Associations between nursing home characteristics and the occurrence of outbreaks ≥3 cases among the residents, adjusted on the occurrence of cases ≥1 among the staff.

Characteristics	Outbreak-Free NH	Outbreak NH	Univariate			Bivariate		
	*n* = 51	*n* = 22	OR	95% CI	*p*-Value	aOR *	95%	*p*-Value
Cases ≥ 1 among the staff	23 (46.0)	19 (90.5)	11.23	2.35–53.63	0.002	-	-	-
NHs’ characteristics								
Presence of coordinating physician	43 (82.7)	20 (90.9)	1.90	0.37–9.80	0.441	-	-	-
High number of residents (>80)	15 (28.8)	7 (31.8)	1.20	0.40–3.57	0.743	0.87	0.26–2.89	0.817
Presence of dementia places	35 (70.0)	16 (72.7)	0.99	0.32–3.08	0.987	0.61	0.16–2.33	0.473
Twin bedrooms ≥ 1	15 (29.4)	7 (31.8)	1.17	0.39–3.47	0.782	0.88	0.26–2.97	0.843
High turnover rate (>2%)	35 (70.0)	13 (59.1)	0.62	0.22–1.76	0.368	0.33	0.09–1.28	0.110
High dependence level (GMP ^a^ score > 80)	24 (52.2)	8 (40.0)	0.61	0.21–1.77	0.365	0.72	0.22–2.37	0.592
High morbidity level (PMP ^b^ score > 225)	23 (54.8)	8 (42.1)	0.60	0.20–1.80	0.362	0.65	0.20–2.16	0.481
High total number of staff (>60)	26 (55.3)	13 (65.0)	1.50	0.51–4.43	0.463	0.93	0.28–3.16	0.911
High number of fulltime caregiver (>40 per 100 residents)	32 (72.7)	10 (62.5)	0.63	0.19–2.10	0.447	0.55	0.15–2.05	0.376
Use of interim manpower	36 (72.0)	13 (59.1)	0.58	0.20–1.65	0.307	0.41	0.11–1.47	0.172
Use of other sources of on-demand manpower	38 (74.5)	19 (86.4)	2.23	0.56–8.77	0.253	1.29	0.28–6.05	0.744
Preventive measures implemented								
Early ban of visits (before the national decision)	15 (32.6)	8 (38.1)	1.23	0.42–3.61	0.706	0.66	0.20–2.24	0.506
Early systematic facial mask wearing (before the national lockdown)	34 (68.0)	15 (68.2)	0.97	0.33–2.88	0.962	1.01	0.30–3.36	0.987
Replacement of the face mask every 4 h	31 (59.6)	13 (59.1)	0.96	0.35–2.67	0.942	1.26	0.40–3.99	0.698
Confining residents’ in their room	42 (82.4)	20 (95.2)	4.50	0.53–38.04	0.167	-	-	-

* The strength of association is adjusted on the occurrence of cases ≥1 among the staff members. ^a^ GMP: “Groupe iso-ressources Moyen Pondéré” is a score assessing the average level of dependency of nursing home residents. The GMP scale is ranging from 70 points (very low level) to 1000 points (very high level). ^b^ PMP: “Pathos Moyen Pondéré”, is a score assessing the average level of morbidity of the nursing home residents. The scale is ranging from 180 (very low level) to 260 (very high level).

## Data Availability

The data can be shared upon request sent to the corresponding author.

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
