# Peer review of "Outbreaks of COVID-19 in Nursing Homes: A Cross-Sectional Survey of 74 Nursing Homes in a French Area"

_jcm, 2021, doi:10.3390/jcm10184280_

Round 1

Reviewer 1 Report

The study results are interesting though predictable. They can inspire  to implement appropriate procedures in subsequent epidemic waves or other similar situations (robotization?, mandatory personnel testing every few days?) Although a brief discussion would also require the observation of more frequent hospitalizations of people who fell ill, in home care conditions. It would  also require reference to  the cause of a greater number of cases in nursing homes in the United States.

"In accordance to French regulation, this study was not approved by an institutional  review board as the data collected were aggregated" better

...did not require ..

Author Response

Comments and Suggestions for Authors

Comment 1: The study results are interesting though predictable. They can inspire to implement appropriate procedures in subsequent epidemic waves or other similar situations (robotization? mandatory personnel testing every few days?)

We improved the discussion regarding potential procedures which may improve the management of future waves or similar situations.         

“This underlines the need to focus […]. In a recent study it was estimated that up to 22% of French healthcare workers were not willing at all to get vaccinated against COVID-19, and that only 30% would be likely to accept a mandatory immunization program [30]. Tools such as the Haddon matrix used in the injury prevention field or the CDC’s Infection Control Assessment and Response (ICAR) tool are thereby of importance both at national and facility levels, to assess feasibility and assist decision making for the prevention and containment of COVID-19 outbreaks in nursing homes [31,32].”

We also updated the method section with a study we conducted was recently published in “Vaccines.”

 “In an unpublished recent survey that we conducted among 3,454 healthcare workers from four public hospitals of the French Alps, the overall prevalence of SARS-CoV-2 infection was 5.0% (95CI, 4.3% to 5.8%) [11].”

Comment 2: Although a brief discussion would also require the observation of more frequent hospitalizations of people who fell ill, in home care conditions.

In the discussion, we added references to two articles published recently in the Lancet Healthy and Longevity. They support the existing relationships between living environment or frailty, and mortality.

  • Residential context and COVID-19 mortality among adults aged 70 years and older in Stockholm: a population-based, observational study using individual-level data
  • Association between Clinical Frailty Scale score and hospital mortality in adult patients with COVID-19 (COMET): an international, multicentre, retrospective, observational cohort study

We also improved the limitations part of the discussion:

“Incidences may have been underestimated as there was no systematic screening, and PCR testing was prompted by the occurrence of symptoms [2,18]. Thus it was not possible to compare the timeline between detection of cases among residents and staff members. Comparison of incidence and mortality in nursing homes and in the elderly population receiving care at home is difficult as the two populations and their environments are not comparable.”

Comment 3: It would also require reference to the cause of a greater number of cases in nursing homes in the United States.

We added a paragraph regarding metrics issues which can arise when comparing outbreaks in nursing homes. The first reference deals with the denominator used in computation of incidences. The second one with the ethnic composition of facilities which was reported as a risk factor for larger and deadlier epidemics in the United States.

“Incidences may have been underestimated as there was no systematic screening, and PCR testing was prompted by the occurrence of symptoms [2,18]. Thus it was not possible to compare the timeline between detection of cases among residents and staff members. Comparison of incidence and mortality in nursing homes and in the elderly population receiving care at home is difficult as the two populations and their environments are not comparable. Also, it was difficult to compare our results with those from other reports as eligibility for care in nursing homes, their organization and level of medical oversight differ greatly within and across countries. Besides, to estimate incidences we used a conservative approach based on the total number of residents present in nursing homes during the surveyed period. Other reports base their estimation on number of occupied bed at baseline, which leads to inflated estimates [33]. We were not able to explore the effect of the ethnicity of facilities personnel and their residents on incidence and mortality, while in the United States ethnicity was associated with the likelihood of COVID-19 cases and death [34].”

Comment 4: "In accordance to French regulation, this study was not approved by an institutional  review board as the data collected were aggregated" better ...did not require ..

Thank you for the remark. We modified accordingly.

“In accordance with French regulations, this study did not require an institutional review board approval as the data collected were aggregated (Article R1121-1 of the French Public Health Code).”

Reviewer 2 Report

Thank you for the review.

The authors are doing a multi-centric cross-sectional survey conducted in nursing homes of the French Alps, to analyze the relationship between the occurrence of an outbreak of COVID 19 among residents and staff members and they found a strong positive correlation was existing between residents and staff members for both the number of cases and the incidence. They concluded that cases among the staff were the only factor associated with the occurrence of an outbreak among residents and this relationship was not influenced by any nursing homes characteristics, nor the action they implemented to mitigate the COVID-19 crisis. Staff members were therefore likely to be a source of contamination and spread of COVID-19 among nursing homes residents during the first wave of the pandemic.

The project is well thought. The methodology is clear and results are presented in a clear way. I would refrain from commenting on statistical analysis as that is not my expertise.

The discussion is well written , and referenced. The language used is appropriate .

I would recommend publication if the statistical analysis is found appropriate.

Thank you

Author Response

The authors are doing a multi-centric cross-sectional survey conducted in nursing homes of the French Alps, to analyze the relationship between the occurrence of an outbreak of COVID 19 among residents and staff members and they found a strong positive correlation was existing between residents and staff members for both the number of cases and the incidence. They concluded that cases among the staff were the only factor associated with the occurrence of an outbreak among residents and this relationship was not influenced by any nursing homes characteristics, nor the action they implemented to mitigate the COVID-19 crisis. Staff members were therefore likely to be a source of contamination and spread of COVID-19 among nursing homes residents during the first wave of the pandemic.

The project is well thought. The methodology is clear and results are presented in a clear way. I would refrain from commenting on statistical analysis as that is not my expertise.

The discussion is well written , and referenced. The language used is appropriate .

I would recommend publication if the statistical analysis is found appropriate.

Thank you for the review.